# Stronger net selection on males across animals

**Lennart Winkler[1], Maria Moiron[2], Edward H Morrow[3], Tim Janicke[1,2]\***

[1]Applied Zoology, Technical University Dresden, Dresden, Germany; [2]CEFE, CNRS, Univ Montpellier, EPHE, IRD, Montpellier, France; [3]Department for Environmental and Life Sciences, Karlstad University, Karlstad, Sweden

**Abstract** Sexual selection is considered the major driver for the evolution of sex differences. However, the eco-evolutionary dynamics of sexual selection and their role for a population's adaptive potential to respond to environmental change have only recently been explored. Theory predicts that sexual selection promotes adaptation at a low demographic cost only if sexual selection is aligned with natural selection and if net selection is stronger on males compared to females. We used a comparative approach to show that net selection is indeed stronger in males and provide preliminary support that this sex bias is associated with sexual selection. Given that both sexes share the vast majority of their genes, our findings corroborate the notion that the genome is often confronted with a more stressful environment when expressed in males. Collectively, our study supports one of the long-standing key assumptions required for sexual selection to bolster adaptation, and sexual selection may therefore enable some species to track environmental change more efficiently.

## Editor's evaluation

This study addresses an interesting and important question in evolutionary biology: how does the variance in fitness (components) vary between the sexes? In particular, it aims to evaluate whether there is a larger sex difference in systems with strong sexual selection. This study will be of considerable interest to researchers working on sexual coevolution and the role of sexual selection in promoting adaptation. However, there are some concerns regarding the limitations of the data and methods in support of the conclusions.

*For correspondence:
tim.janicke@cefe.cnrs.fr

**Competing interest:** The authors declare that no competing interests exist.

## Introduction

For almost a century, researchers have gathered compelling evidence that sexual selection (i.e., selection arising from competition for mating partners and/or their gametes) constitutes the ultimate evolutionary force generating sexual dimorphism in a multitude of reproductive characters and life-history traits (*Andersson, 1994*; *Clutton-Brock, 2007*). Despite this progress, we are just beginning to understand the eco-evolutionary dynamics of sexual selection in terms of its impact on demography and adaptive potential of a population (*Svensson and Deere, 2018*). On the one hand, sexual selection has often been considered to promote the evolution of traits that have opposing fitness effects in both sexes, which manifests in sexual conflict (*Arnqvist and Rowe, 2005*). Such sexually antagonistic selection is predicted to have a demographic cost and therefore to reduce the population's adaptive potential (*Chapman, 2006*; *Rankin et al., 2011*). More precisely, sexual selection may lead to interlocus sexual conflict by favoring the evolution of traits that are beneficial in one sex (also called persistence traits such as harassment and infanticide) but induce a fitness cost in the other sex and thereby promote the evolution of traits that mitigate these costs (resistance traits) (*Arnqvist and*

*Rowe, 2005*). Furthermore, if the expression of a shared trait has different fitness consequences for males and females, intralocus sexual conflict may shift male and female phenotypes away from their evolutionary optimum. In particular, intralocus sexual conflict for fitness implies that sexual selection in one sex favors genotypes that have a low reproductive fitness when expressed in the other, which may manifest in negative cross-sex genetic correlations of fitness (*Chippindale et al., 2001*; *Foerster et al., 2007*) and thereby lower the population's adaptive potential (*Pischedda and Chippindale, 2006*; *Bonduriansky and Chenoweth, 2009*).

In contrast to the predictions obtained from sexual conflict thinking, another line of theoretical and empirical work suggests that sexual selection can facilitate how populations cope with environmental change (*Lorch et al., 2003*; *Candolin and Heuschele, 2008*; *Holman and Kokko, 2013*; *Martínez-Ruiz and Knell, 2017*; *Martinossi-Allibert et al., 2019*; *Rowe and Rundle, 2021*). This latter school of thought is based on two main assumptions. First, we need to assume that intralocus sexual conflict is rare so that cross-sex genetic correlations of fitness are positive (*Rowe and Rundle, 2021*). This implies that sexual and natural selection are aligned, meaning that sexual selection favors alleles that also improve fecundity and survival – a process mediated by condition dependence of sexually selected traits (*Rowe and Houle, 1997*). Such an alignment is expected to manifest in a positive genetic correlation between male and female fitness as long as sexually antagonistic loci are rare and/or have minor fitness effects. *Poissant et al., 2010* compiled cross-sex genetic correlations of various trait types and found that correlations are generally large and positive but tend to be lower for fitness components in comparison to morphological, behavioral, and developmental traits. Moreover, there is strong comparative and meta-analytic evidence supporting the idea that the expression of pre- and postcopulatory sexual traits depends on the overall condition of the male (*Cotton et al., 2004*; *Macartney et al., 2019*). This means that sexual selection may not only favor the evolution of prominent secondary sexual traits but also traits that confer health and vigor (*Jennions et al., 2001*), and therefore eventually purges deleterious alleles that are targeted by natural selection.

In contrast to the extensive support for an alignment of sexual and natural selection in many species (*Whitlock and Agrawal, 2009*; *Cally et al., 2019*; *Rowe and Rundle, 2021*), we still know very little on whether the second key assumption for sexual selection to empower evolutionary adaptation is generally fulfilled across a broad range of taxa. Namely, sexual selection enforces natural selection only if it gives rise to stronger net selection (defined as the sum of genome-wide selection against deleterious alleles) on males compared to females. In a landmark synthesis paper, *Whitlock and Agrawal, 2009* explored the effect of sex-specific selection on the population's mutation load, that is the overall reduction of absolute fitness due to deleterious alleles in a population. They expanded the foundational work of *Haldane, 1937* by relaxing the assumption of random mating to demonstrate that the mutation load $L$ is expected to be

$$L = 2\mu \left( \frac{\frac{s_f}{s}}{s} \right)$$

where $\mu$ is the mutation rate from wild type to deleterious alleles and $s$ is the average selection coefficient against deleterious alleles of females ($s_f$) and males. Hence, the mutation load is reduced whenever $s_f < s$, which arises if net selection is stronger on males compared to females. Given that the population's productivity is typically governed by female fecundity, a population with stronger net selection on males can purge its mutation load and adapt faster to a new environment with a lowered demographic cost and thereby reduce its extinction risk. In other words, females benefit from being part of a gene pool that is purified primarily through stronger selection on males (*Whitlock and Agrawal, 2009*). It is important to note that this perspective assumes that the vast majority of deleterious alleles have negative fitness effects on both male and female fitness, which implies that sex-limited genes are relatively rare in the genome.

There is ample empirical evidence that typically (though not always) males undergo stronger sexual selection whereas females are primarily exposed to fecundity selection (*Janicke et al., 2016*) as predicted by Bateman's principle (*Bateman, 1948*). However, our knowledge on whether such stronger sexual selection on males eventually translates into stronger net selection relative to females is still limited and equivocal (*Whitlock and Agrawal, 2009*; *Hendry et al., 2018*). This lack of evidence stems at least partially from difficulties in quantifying the strength of net selection in a framework that also allows comparisons among sexes and species. Potentially the most promising metric to contrast the strength of net selection across contexts is the mean-standardized variance in fitness,

which is often expressed as the coefficient of variation (*CV*). Previous comparative studies provided evidence that the phenotypic variance in fitness ('opportunity for selection' [*Crow, 1958*]) is typically larger in males compared to females, suggesting that the upper opportunity for net selection is stronger in males (*Janicke et al., 2016*). However, it has been questioned whether the phenotypic variance in fitness provides a good proxy for the strength of net selection because environmental variation can substantially inflate this variance, which limits its explanatory power with respect to evolutionary responses and complicates the comparison across contexts and studies (*Whitlock and Agrawal, 2009*). Moreover, male reproductive success is often quantified with higher uncertainty due to measurement error during paternity analysis, which may lead to higher phenotypic variance in fitness of males compared to females caused by sex-specific methodology rather than by a biological mechanism. To overcome this problem, researchers have advocated to use the genetic rather than the phenotypic variance in fitness as a proxy for the strength of net selection (*Jones, 2009*; *Hendry et al., 2018*). This is also because the genetic variance in relative fitness corresponds to the rate of increase in mean fitness that results from selection on allele frequencies (*Fisher, 1930*). Therefore, the mean-standardized genetic variance (*CV*$_G$) in fitness provides a highly diagnostic metric to quantify the strength of net selection, but it has to our knowledge not yet been subjected to a systematic and global test for an overall sex difference.

Here, we addressed this key aspect of sexual selection theory by using a comparative approach to test whether net selection is generally stronger on males across a broad taxonomic range. Specifically, we ran a systematic literature search and compiled 101 pairwise estimates of male and female genetic variances for 2 main fitness components – reproductive success and lifespan – from a total of 26 species. Applying phylogenetically informed comparative analyses we tested (1) whether phenotypic variances are aligned to genetic variances as assumed by the phenotypic gambit (*Grafen, 1991*) and (2) whether males show larger genetic variance in reproductive success but not lifespan. According to the so-called *genic capture hypothesis*, an organism's condition (defined as the pool of acquired resources that can be allocated into different fitness components such as survival and reproductive success) is a quantitative trait governed by a large array of loci and determines the fitness of an organism by affecting its survival, fecundity, and the outcome of pre- and postcopulatory sexual selection (*Rowe and Houle, 1997*). Importantly, if sexual selection operates on males, their reproductive success not only depends on gamete production and postzygotic investment (as predicted to constitute the primary determinants in females), but also on the outcome of pre- and postcopulatory mate choice and mate competition. Therefore, a deleterious allele with its negative effect on condition is

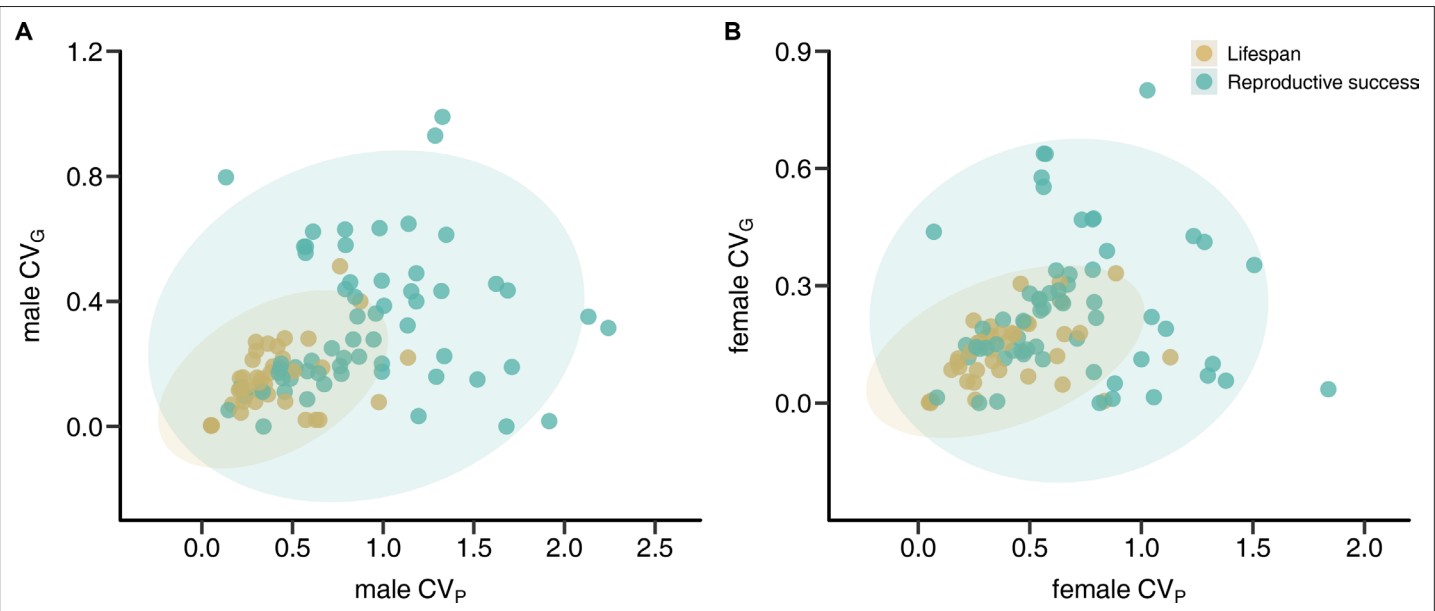

**Figure 1.** Correlations between phenotypic and genetic coefficients of variation (*CV*$_P$ and *CV*$_G$, respectively) for lifespan and reproductive success. Scatterplots show relationships between *CV*$_P$ and *CV*$_G$ for male (**A**) and female (**B**) reproductive success (green) and lifespan (brown). Shaded areas indicate the 95 % confidence ellipses.

predicted to have a disproportionately higher impact on reproductive success in males compared to females. This heightened condition dependence of male performance (*Rowe and Houle, 1997*) is expected to translate ultimately into a higher genetic variance in reproductive success in males (*Whitlock and Agrawal, 2009*; *Hendry et al., 2018*). By contrast, lifespan is considered to be unaffected by the outcome of sexual selection except in the relatively rare event of mortal combats for access to mates, mate harassment or indirectly by shaping life-history strategies (*Bondurianský et al., 2008*). If correct, a deleterious allele is therefore expected to have a similar effect on lifespan in males and females so that we predict if anything, a much weaker male bias for genetic variance in lifespan. In this context, we also provide a preliminary test on the role of sexual selection for generating the hypothesized sex differences in net selection. Specifically, we contrasted socially monogamous and polygamous species, with the prediction that a male bias in genetic variance for reproductive success is primarily prevalent in polygamous species where sexual selection is most likely to be stronger compared to monogamous species (*Shuster and Wade, 2003*).

## Results

We found that the phenotypic coefficient of variation ($CV_P$) of reproductive success does not predict the genetic coefficient of variation ($CV_G$) in either males (linear regression: estimate ± SE = 0.085 ± 0.060, $F_{1,60}$ = 2.026, p = 0.160, $R^2$ = 0.03) or females (linear regression: estimate ± SE = 0.038 ± 0.064, $F_{1,60}$ = 0.350, p = 0.667, $R^2$ = 0.01; *Figure 1*). Similarly, phylogenetic independent contrasts (PICs) analyses suggest only weak though statistically significant relationships between $CV_P$ and $CV_G$ for reproductive success in males (linear regression: estimate ± SE = 0.113 ± 0.051, $F_{1,37}$ = 4.847, p = 0.032, $R^2$ = 0.07) and females (linear regression: estimate ± SE = 0.191 ± 0.065, $F_{1,37}$ = 8.615, p = 0.005, $R^2$ = 0.13). In contrast, $CV_P$ was a strong predictor of $CV_G$ for lifespan in both males (linear regression: estimate ± SE = 0.266 ± 0.068, $F_{1,60}$ = 15.3, p < 0.001, $R^2$ = 0.29) and females (linear regression: estimate ± SE = 0.204 ± 0.051, $F_{1,60}$ = 16.16, p < 0.001, $R^2$ = 0.30). These effects were stronger when accounting

**Table 1.** Results of phylogenetic general linear mixed-effect models testing for an effect of sex on phenotypic ($CV_P$) and genetic ($CV_G$) coefficient of variation.

Results shown for reproductive success (RS) and lifespan (LS) for models ran across mating systems and when ran separately for socially monogamous and polygamous species. Estimates are shown as posterior means with 95 % highest posterior density (HPD) intervals, with positive values indicating a male bias. $P_{MCMC}$ is the probability of the posteriors including zero. The variance explained by sex is given as the marginal $R^2$ and the phylogenetic signal is reported as $H^2$.

| Response | Variance component | Sex effect estimate | | $P_{MCMC}$ | Marginal $R^2$ | | | Phylogenetic $H^2$ | | |
|---|---|---|---|---|---|---|---|---|---|---|
| *Across mating systems* | | | | | | | | | | |
| RS | $CV_P$ | 0.234 | (0.149 ,0.322) | <0.001 | 0.07 | (0.02, | 0.12) | 0.12 | (0.00, | 0.32) |
| | $CV_G$ | 0.086 | (0.043, 0.128) | <0.001 | 0.04 | (0.00, | 0.08) | 0.28 | (0.03, | 0.58) |
| LS | $CV_P$ | −0.005 | (−0.031, 0.021) | 0.704 | 0.00 | (0.00, | 0.00) | 0.41 | (0.06, | 0.81) |
| | $CV_G$ | 0.017 | (−0.006, 0.040) | 0.149 | 0.01 | (0.00, | 0.02) | 0.46 | (0.12, | 0.81) |
| *Monogamy* | | | | | | | | | | |
| RS | $CV_P$ | −0.012 | (−0.063, 0.035) | 0.602 | 0.00 | (0.00, | 0.00) | 0.37 | (0.01, | 0.96) |
| | $CV_G$ | −0.015 | (−0.080, 0.046) | 0.628 | 0.01 | (0.00, | 0.02) | 0.43 | (0.05, | 0.92) |
| LS | $CV_P$ | 0.026 | (−0.028, 0.079) | 0.281 | 0.00 | (0.00, | 0.02) | 0.37 | (0.01, | 0.87) |
| | $CV_G$ | 0.050 | (−0.030, 0.125) | 0.179 | 0.02 | (0.00, | 0.07) | 0.46 | (0.05, | 0.89) |
| *Polygamy* | | | | | | | | | | |
| RS | $CV_P$ | 0.312 | (0.207, 0.417) | <0.001 | 0.11 | (0.03, | 0.18) | 0.20 | (0.01, | 0.47) |
| | $CV_G$ | 0.119 | (0.068, 0.170) | <0.001 | 0.07 | (0.01, | 0.13) | 0.25 | (0.02, | 0.56) |
| LS | $CV_P$ | −0.014 | (−0.046, 0.018) | 0.373 | 0.00 | (0.00, | 0.00) | 0.69 | (0.25, | 0.97) |
| | $CV_G$ | 0.007 | (−0.016, 0.031) | 0.554 | 0.00 | (0.00, | 0.01) | 0.44 | (0.12, | 0.78) |

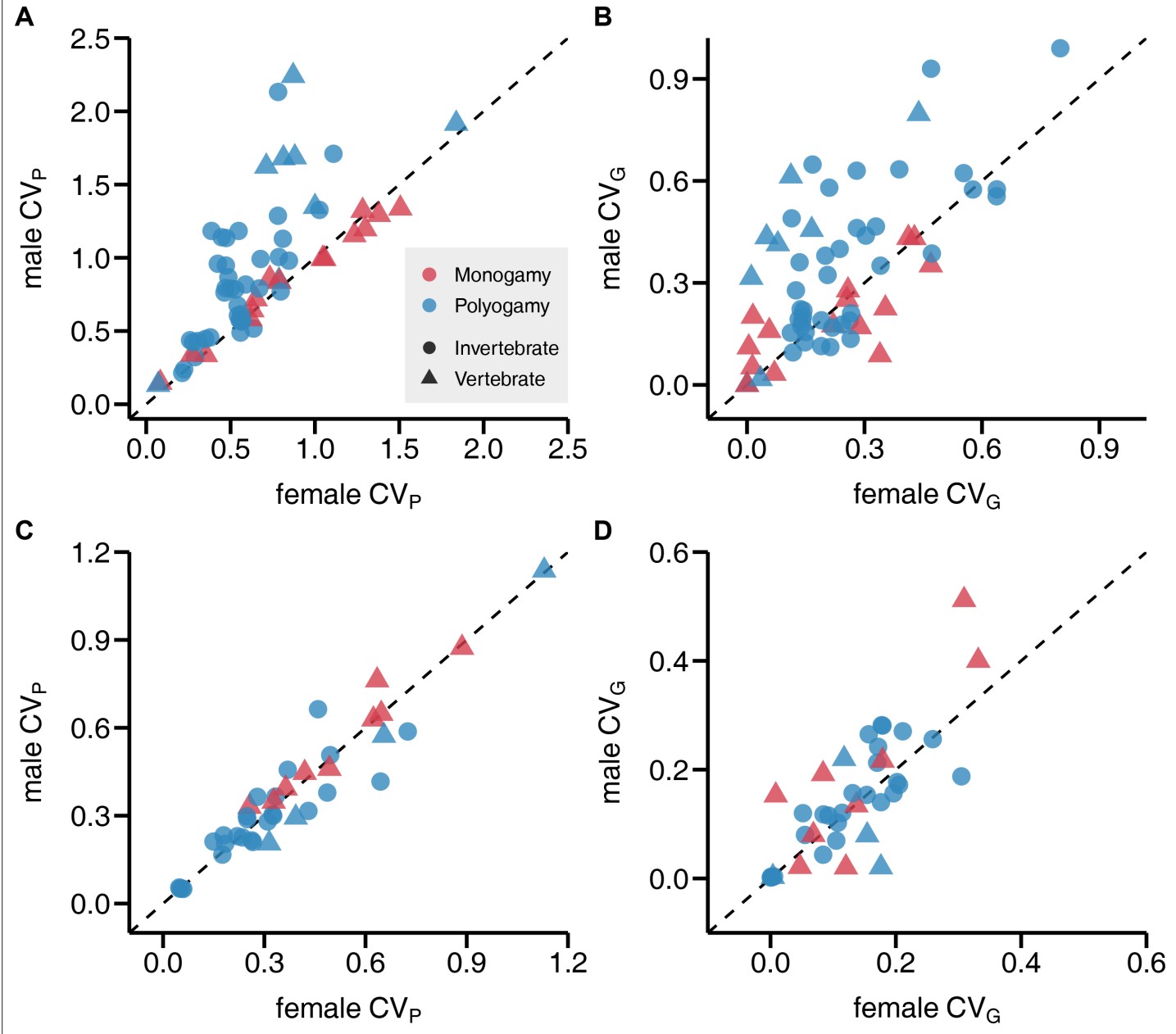

**Figure 2.** Sex bias in phenotypic and genetic variances in reproductive success and lifespan. Scatterplots show the coefficient of phenotypic variation $CV_P$ (**A, C**) and genetic variation $CV_G$ (**B, D**) for reproductive success (**A, B**) and lifespan (**C, D**). Monogamous species are represented in red, polygamous species in blue. All data points above the diagonals indicate a male bias.

for phylogenetic nonindependence in males (linear regression: estimate ± SE = 0.552 ± 0.048, $F_{1,37}$ = 131.1, p < 0.001, $R^2$ = 0.78) and females (linear regression: estimate ± SE = 0.345 ± 0.036, $F_{1,37}$ = 91.14, p < 0.001, $R^2$ = 0.70).

Most importantly, $CV_P$ of reproductive success was generally larger in males compared to females, which translated into a male bias in $CV_G$, with sex explaining 7 % and 4 % of the observed variance, respectively (*Table 1*; *Figure 2A-B* and *Figure 3*). Interestingly, when running a preliminary test, we found that this sex difference could be detected in polygamous but not monogamous species, which manifested in a significant sex by mating system interaction (*Table 1* and *Supplementary file 1*; *Figure 3*). These findings could be confirmed when analyses were run on a subset including only vertebrate species (*Supplementary files 2 and 3*). Contrary to the results for reproductive success, we did not observe consistent sex differences in $CV_P$ and $CV_G$ for lifespan (*Table 1*; *Figure 2C-D*). Finally,

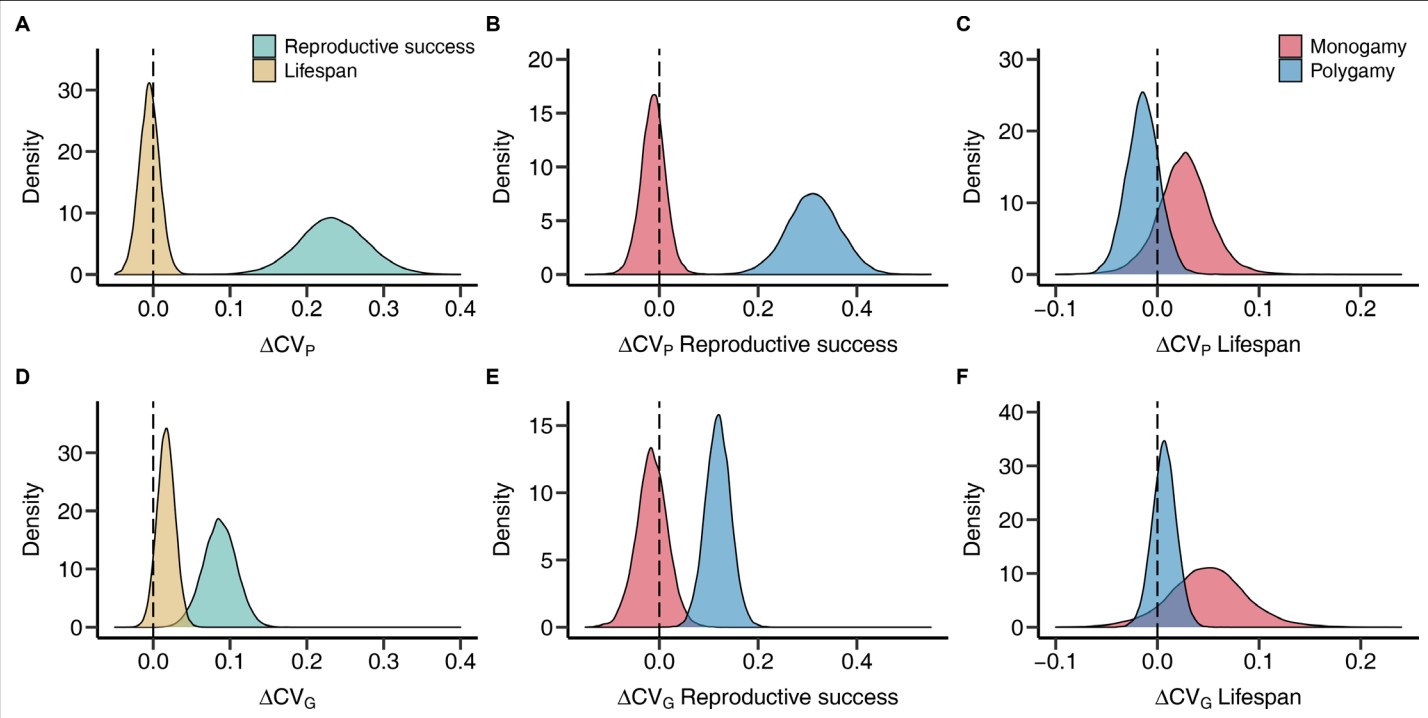

**Figure 3.** Sex differences in phenotypic and genetic coefficients of variation for reproductive success and lifespan. Plots show posterior distributions for the sex difference of the phenotypic (**A–C**) and genetic (**D–F**) coefficient of variation ($\Delta CV_P$ and $\Delta CV_G$, respectively) obtained from phylogenetic general linear mixed-effects models (PGLMMs; see Methods). Positive values indicate a male, negative values a female bias. Density plots contrast fitness components pooled across mating systems (**A and D**) or compare socially monogamous and polygamous species separately for reproductive success (**B and E**) and lifespan (**C and F**).

methodological heterogeneity among primary studies assessed in terms of study type, the estimate of genetic variance $V_G$, and the type of reproductive success metric did not predict $CV_P$ and $CV_G$ or the observed sex differences (***Supplementary files 4–6***).

## Discussion

Males and females share the vast majority of their genome but are often subject to fundamentally different selection pressures, which is predicted to impact the demography and the adaptive potential of a population when facing environmental change (***Whitlock and Agrawal, 2009***; ***Holman and Kokko, 2013***; ***Svensson and Deere, 2018***). In line with sexual selection theory, our study provides the first comparative evidence that genome-wide selection is generally stronger on males compared to females. More specifically, our results have two major implications. First, phenotypic variance is a good predictor of genetic variance for lifespan but not for reproductive success, providing limited support for the phenotypic gambit. Therefore, the opportunity for selection measured as the phenotypic variance in reproductive success is a poor proxy for the strength of net selection. Despite these results, our second major finding is that the previously observed male bias in the phenotypic opportunity for selection (***Janicke et al., 2016***) is also reflected in an overall higher male genetic variance in reproductive success. Interestingly, our preliminary analysis of the mating system revealed that this male bias can only be detected in polygamous species, in which sexual selection is likely to be stronger compared to monogamous species (***Shuster and Wade, 2003***). Hence, pre- and postcopulatory competition and/or mate choice may magnify the material that selection acts on in a sex-specific manner. In contrast, phenotypic and genetic variances of lifespan do not show sex differences in either polygamous or monogamous species. The sample size for lifespan ($N_{Estimates}$ = 39; $N_{Species}$ = 16) was smaller compared to reproductive success ($N_{Estimates}$ = 62; $N_{Species}$ = 21) meaning that we had less statistical power to detect a sex difference in phenotypic and genetic variances for lifespan. However, we did not observe any consistent trend for lifespan and our data suggest that even if there was a

sex difference in genetic variance in lifespan, its effect size would be considerably smaller compared to reproductive success. Thus, we conclude that while sexual selection may promote sex differences in mean lifespan at an evolutionary scale (*Lemaître et al., 2020*), our results indicate that it does not generate sex-specific genetic variances in this fitness component. Hence, the strength of selection on survival appears to be similar among males and females.

Despite the overall strong sex difference in genetic variation of reproductive success, our results on the effect of the social mating system need to be considered with caution. This is because of the underrepresentation of monogamous species in our dataset and because of only three independent evolutionary changes from polygamy to monogamy in our phylogeny. Moreover, our binary classification of the mating system into socially monogamous and polygamous species fails to capture the continuum in the strength of sexual selection across the sampled taxa, likely limiting its explanatory power as a predictor variable. Ideally, one would use a standardized continuous estimate for the strength of sexual selection allowing inter- and intraspecific comparisons such as Bateman metrics, for example, the maximum intensity of precopulatory sexual selection $s'_{max}$ (*Jones, 2009*; *Henshaw et al., 2016*). Unfortunately, such data are currently only accessible for a small fraction of the sampled species (i.e., 19 % based on a database collected by *Janicke and Morrow, 2018*), which renders those measures as additional predictors for the strength of sexual selection unavailable at the current state of knowledge.

In essence, our findings provide support for the prediction that net selection is stronger on males compared to females. We conclude that one of the key assumptions required for sexual selection to assist natural selection and thereby to accelerate the adaptation to changing environments is often fulfilled in nature. For species with a positive cross-sex genetic correlation of fitness (i.e., limited intralocus sexual conflict), stronger net selection on males implies that populations may purge deleterious alleles across the genome primarily at the expense of males and thus at a low demographic cost (*Agrawal, 2001*; *Siller, 2001*). This has important eco-evolutionary consequences because stronger net selection on males will not only bolster local adaptation but will also reduce extinction risk when populations are coping with challenging environmental conditions (*Lumley et al., 2015*). Therefore, our findings support the idea that sexual selection can play a pivotal role in evolutionary rescue (*Candolin and Heuschele, 2008*; *Holman and Kokko, 2013*; *Svensson and Deere, 2018*) and are in line with a recent meta-analysis providing compelling evidence that sexual selection increases nonsexual fitness (*Cally et al., 2019*). Interestingly, environmental stress has repeatedly been found to elevate the effect of deleterious mutations and thereby increase genetic variation in fitness-related traits (*Rowiński and Rogell, 2017*). Thus, we extend the sexes-as-environments analogy (*Rice and Chippindale, 2008*) to say that an almost identical genome is expressed in a more stressful male environment versus a relatively more benign female environment.

Whether stronger net selection on males eventually promotes adaptation to a new environment, or even contributes to the evolution and maintenance of sexual over asexual reproduction (*Agrawal, 2001*; *Siller, 2001*), will also depend on whether sexual conflict impedes the alignment of natural and sexual selection. This includes another important aspect of the genetic architecture of male and female fitness components: intralocus sexual conflict and its impact on the cross-sex genetic covariance of fitness. Specifically, only if fitness in both sexes is condition dependent (i.e., positively affected by the amount of acquired resources) and therefore largely governed by a similar set of genes, will sexual selection on males purge deleterious alleles in females and thereby facilitate adaptation (*Whitlock and Agrawal, 2009*). Theoretical work predicts that cross-sex genetic correlations for fitness and fitness components are often low but positive (*Connallon and Matthews, 2019*). Negative genetic correlations between sexes are expected to be rare and only to prevail if two conditions are met: selection is highly antagonistic and that both sexes are strongly displaced from their optimum (*Connallon and Matthews, 2019*). This seems to be in line with empirical work, because studies reporting positive cross-sex genetic correlations outnumber those with negative correlations (*Poissant et al., 2010*; *Connallon and Matthews, 2019*; but see *Foerster et al., 2007*). Only a small fraction of the primary studies included in our analysis reported cross-sex genetic correlations, but exploratory analyses of this subset (*Supplementary file 7*) support an earlier finding of highly positive genetic correlations for lifespan with no consistent pattern for reproductive success (*Hendry et al., 2018*). In the context of adaptation, it is also crucial to understand how environmental change affects cross-sex genetic correlations. Theory predicts that directional selection erodes genetic variation at

loci with concordant effects on males and females more efficiently compared to sexually antagonistic loci. As a consequence, sexually antagonistic loci may have a higher impact on fitness components in well-adapted populations compared to populations facing novel environmental conditions (*Connallon and Hall, 2016*). Therefore, we may expect that environmental changes are generally associated with more positive cross-sex genetic correlations for fitness components, but the few experimental studies provided rather mixed empirical support for this prediction (*Delcourt et al., 2009*; *Berger et al., 2014*; *Holman and Jacomb, 2017*). Further work on the genetic covariance between male and female fitness components is clearly needed to evaluate the overall potential of sexual selection to facilitate or constrain the adaptation to changing environments. Specifically, how genetic variances and covariances of male and female fitness components change with ecological conditions is largely unknown, though such knowledge is crucial to predict evolutionary trajectories when populations face environmental change. Moreover, for some taxa, sexual selection has been found to increase extinction risk (*Doherty et al., 2003*; *Le Galliard et al., 2005*) potentially as a consequence of intense sexual conflict but the quantitative genetics of male and female fitness of those species remain mostly unexplored.

Despite the detected sex differences and the effect of the social mating system, a large fraction of the intra- and interspecific variance in $CV_G$ remained unexplained. This is potentially, at least in part, because genetic variances are often estimated with low precision, which may have introduced substantial noise into our analyses. Besides that, we suspect that another part of the unexplained variation stems from environmental effects influencing genetic (co)variances (i.e., genotype by environmental interactions) (*Rowiński and Rogell, 2017*), which limits comparisons of studies conducted under different experimental conditions. Moreover, there are also methodological differences between primary studies, which may have contributed to the unexplained variation in $CV_G$. This includes the application of different breeding designs used to quantify genetic variances (i.e., pedigrees, half-sib/full-sib breeding, parent-offspring regressions, and inbred lines), differences in the analytical approaches (latent-scale versus data-scale estimates of genetic variances), and disparity in the measurement of reproductive success (annual versus lifetime reproductive success; see Material and methods). While all these sources of uncontrolled variation are likely to have introduced noise into our dataset, we are not aware of any systematic biases that they might have created. Finally, our study covers a broad taxonomic range spanning flatworms, mollusks, arthropods, and vertebrates but is based on relatively few species. This is admittedly a limitation of our study but at the same time illustrates the clear need for more quantitative genetic studies measuring genetic variation of fitness components in both sexes.

Collectively, our analysis reveals a pervasive male bias in the strength of net selection and provides preliminary support for the role of sexual selection to promote this sex difference. However, we have just started to understand the eco-evolutionary consequences of sexual selection in terms of its impact on demography and the adaptive potential of populations to cope with changing environments.

## Materials and methods
### Literature search and characterization of primary studies

We ran a systematic literature search in order to obtain an unbiased sample of coefficients of genetic variation for two major fitness components: reproductive success and lifespan. Even though we expects that both components contribute largely to overall fitness of an organism, they may differ in their explanatory power. Specifically, reproductive success (i.e., number of offspring or grand-offspring) likely translates proportionally into fitness whereas lifespan may not necessarily show a similar linear relationship (e.g., due to senescence). Therefore, genetic variance in reproductive success may represent a better predictor for net selection than lifespan.

We screened the ISI Web of Science Core Collection database (Clarivate Analytics) on August 2, 2019 (*Supplementary file 8*). This search yielded 3793 records. Moreover, we screened previous synthesis articles on related questions (*Poissant et al., 2010*; *Hendry et al., 2018*; *Connallon and Matthews, 2019*) for other primary studies by which we could identify one additional record (*Wheelwright et al., 2014*). Furthermore, we posted a request on 'evoldir' mailing list (http://life.mcmaster.ca/evoldir.html) and ResearchGate platform (https://www.researchgate.net/) for unpublished data, which resulted in one additional study (Jessica Abbott and Anna Norden, unpublished data). Finally, we added two published studies indicated by colleagues (*Gay et al., 2011*; *Pélissié et al., 2012*)

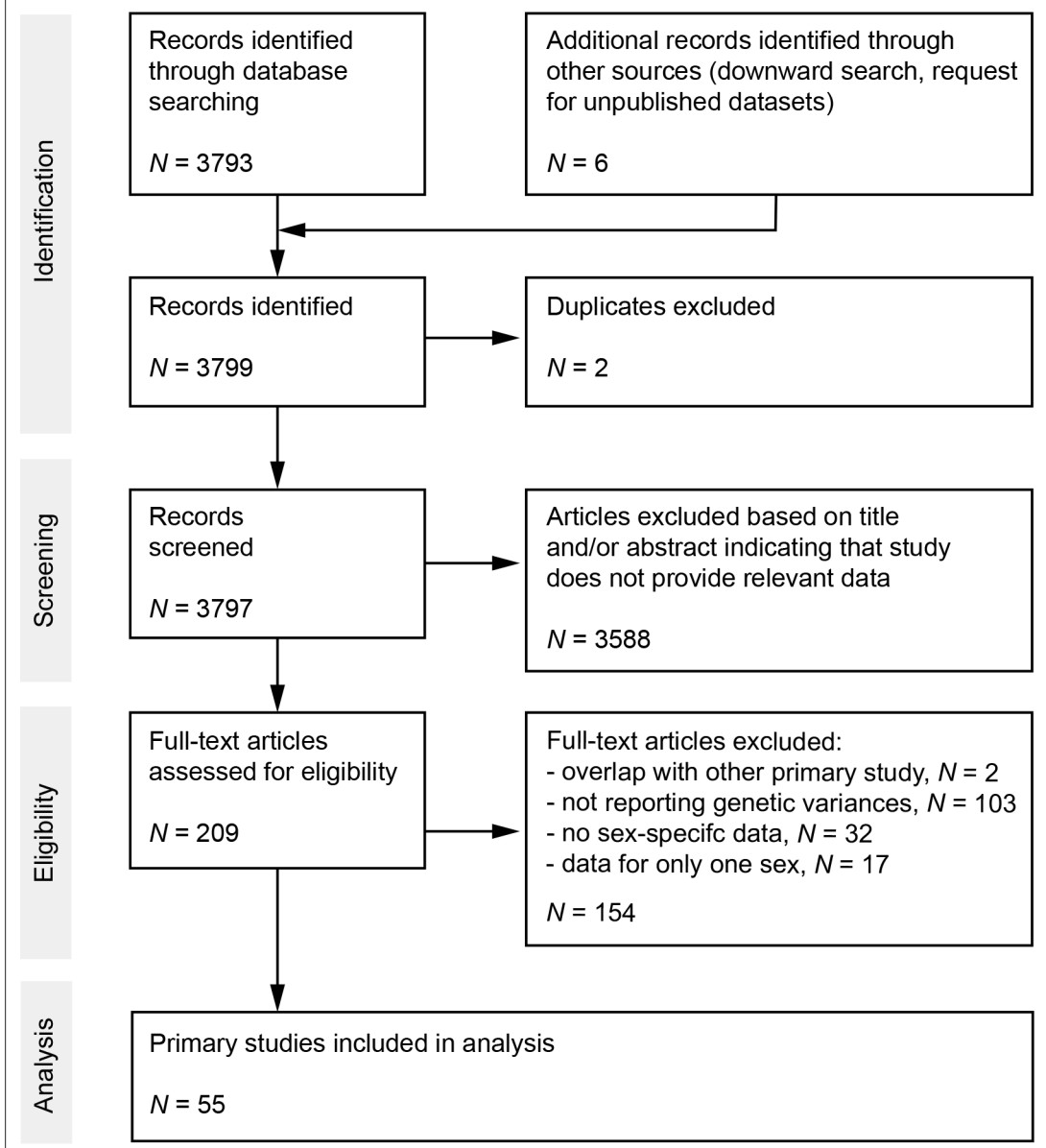

**Figure 4.** Preferred Reporting Items for Systematic Reviews and Meta-Analyses (PRISMA) diagram. Flow chart maps the number of records identified during the different phases of the systematic literature search.

and two of our own studies (*Janicke et al., 2021*.; Maria Moiron, unpublished data). After exclusion of duplicates and screening of abstracts, we checked a total of 203 records for eligibility based on 3 selection criteria. First, studies must report or include information to compute coefficients of genetic variation of reproductive success and/or lifespan. Second, studies must report genetic parameters for males and females both quantified under same conditions (i.e., same field populations or same experimental laboratory conditions). Third, we only included studies on animals simply because of the scarcity of data on genetic variances of male and female fitness components in plants. The final dataset included data from 55 primary studies (see Supporting Information for Preferred Reporting Items for Systematic Reviews and Meta-Analyses [PRISMA] diagram [*Figure 4*] and reference list of all primary studies).

Primary studies varied largely in terms of several methodological aspects (*Supplementary file 9*). First, our dataset includes 23 field studies and 32 laboratory studies. Moreover, primary studies differed in terms of how genetic variances were estimated including full-sib breeding designs (3 studies), half-sib breeding designs (12 studies), inbred lines (17 studies), pedigrees (21 studies), and

twin studies (2 studies). This translated into differences regarding the obtained estimates of genetic variance with 33 studies reporting estimates of additive genetic variance and 22 studies reporting estimates of total genetic variance including additive variance, dominance variance, and epistatic variance. In addition, reproductive success was measured either for a limited period (22 studies) or in terms of lifetime reproductive success (20 studies). Reproductive success also varied with respect to the measured unit such that 24 studies quantified the number of adult offspring, 15 studies the number of juvenile offspring, 1 study the number of grand-offspring, and 2 studies quantified reproductive success in a sex-specific manner. Lifespan was primarily measured as adult survival (i.e., excluding mortality until reaching maturity; 33 out of 39 estimates) and in the few remaining cases as the age of last reproduction, reproductive lifespan (calculated as time between first and last reproduction), or total lifespan (including juvenile mortality). Finally, primary studies differed in the way that male reproductive success was estimated because paternity assessment is required in experimental setups that allow for male–male competition. Specifically, paternity was assessed based on social parentage (14 studies), phenotyping offspring using a genetic maker (i.e., using mutant lines as competitors; 13 studies), genotypic offspring using molecular techniques (5 studies), genotyping offspring in combination with behavioral observations (4 studies), and using the sterile male technique (i.e., sterilizing male competitors with radiation; 3 studies). In three studies, paternity assessment was not required because the experimental setup did not allow for male–male competition (i.e., enforced monogamy). This methodological heterogeneity may have introduced potential biases in absolute variance estimates but we do not expect that the different experimental approaches led to a systematic bias in the sex difference of genetic variance for at least two reasons. First, 53 out of the 55 primary studies (96%) used the same metric to quantify fitness components in both sexes (*Supplementary file 9*), suggesting that potential sex biases due to sex-specific measurements of fitness can only make a very minor contribution to the overall test of sex-specific genetic variances. Second, paternity is often assessed with a higher degree of uncertainty compared to maternity (e.g., due to incomplete and erroneous genotyping of offspring or due to binomial sampling variance), which may result in a higher variance in reproductive success of males compared to females. However, this sex-specific uncertainty in measuring reproductive success might lead to a male bias in phenotypic variance but not in genetic variance because the former is computed from the raw data (including uncontrolled residual variance) whereas the latter is the estimated variance that can attributed exclusively to genetic effects.

## Data acquisition

For all primary studies, we extracted four parameters for both sexes: (1) sample size, (2) arithmetic mean, (3) phenotypic variance, and (4) genetic variance of reproductive success and/or lifespan. For 11 studies, at least one of these parameters was not reported in the article. In these cases, we received the parameter estimates from the authors upon request or reanalyzed the raw data (either published together with the article or provided by the authors). We computed the coefficients of phenotypic and genetic variation ($CV_P$ and $CV_G$, respectively) as the square root of the variance (i.e., the standard deviation) divided by the arithmetic mean, which makes this metric comparable across contexts and species. Note that $CV$ of a given trait is often denoted as 'evolvability' (*Houle, 1992*) and equals the square root of the opportunity for selection $I$, which is also frequently used to quantify the upper limit of the strength of selection (*Jones, 2009*; *Hendry et al., 2018*). In total, we obtained 101 paired estimates of $CV_P$ and $CV_G$ for males and females, including 62 estimates for reproductive success and 39 estimates for lifespan.

## Phylogeny and mating system classification

The 55 primary studies encompass a total of 26 animal species with an overrepresentation of insects ($N = 12$) and birds ($N = 7$). In order to account for any source of phylogenetic nonindependence we reconstructed the phylogeny of all sampled species (*Figure 5*). First, we retrieved pairwise estimates of divergence times from the TimeTree database (http://www.timetree.org/; *Kumar et al., 2017*). Second, we aged undated nodes on the basis of divergence times of neighboring nodes applying the branch length adjuster (BLADJ) algorithm (*Webb et al., 2008*). Finally, we used the resulting distance matrix to compute a phylogeny, using the unweighted pair group method with arithmetic mean (UPGMA) algorithm implemented in MEGA (https://www.megasoftware.net/; *Kumar et al., 2018*) and transformed it into the Newick format for further analysis.

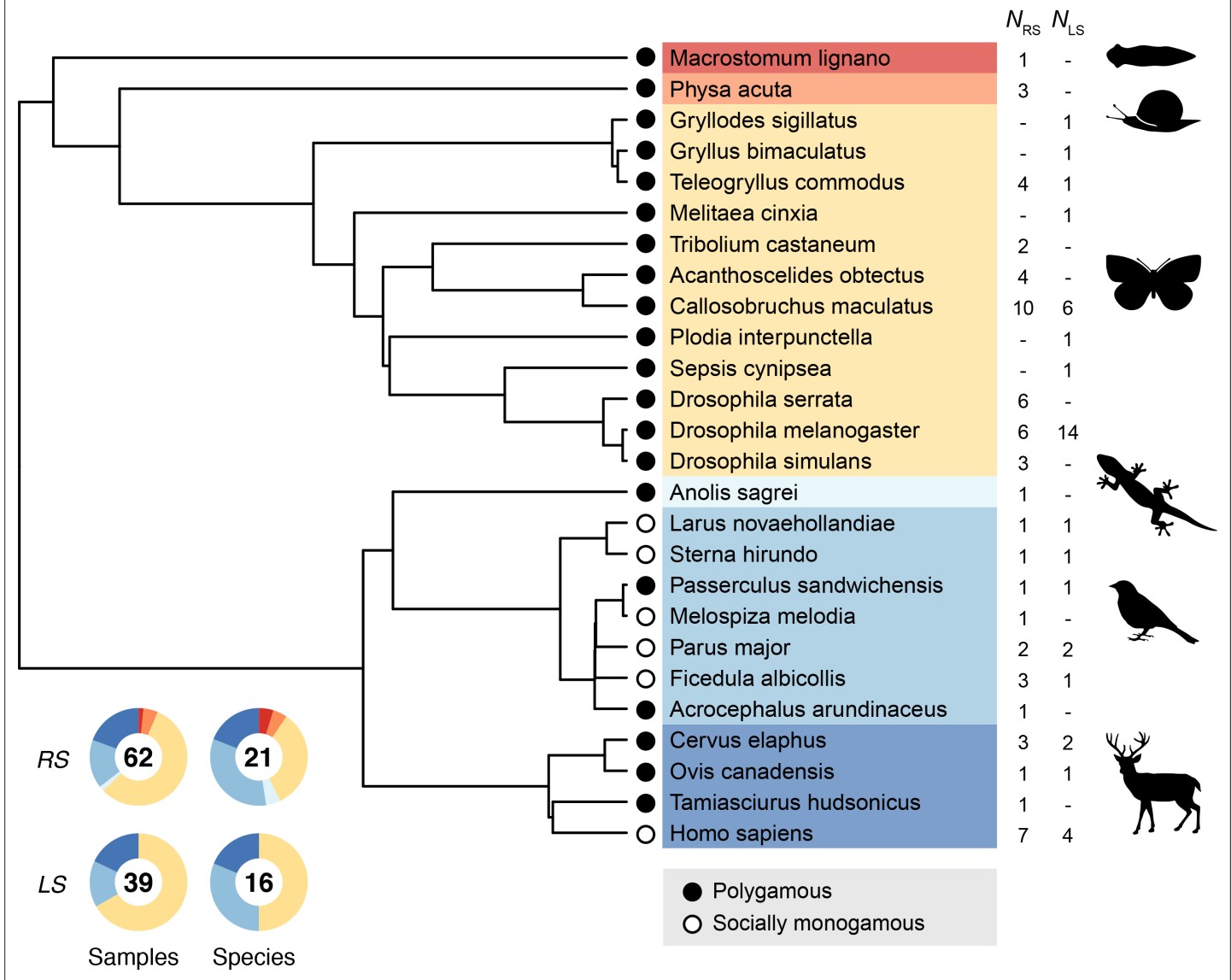

**Figure 5.** Phylogeny used to account for phylogenetic nonindependence in statistical modeling. Doughnut charts show the relative fraction of samples (i.e., number of paired estimates for male and female genetic variance) and the number of species for reproductive success (RS) and lifespan (LS).

To explore the role of sexual selection in generating sex differences in phenotypic and genetic variances, we used published information in the scientific literature on the social mating system as a proxy for the strength of sexual selection. Specifically, we distinguished between socially monogamous ($N = 6$) and polygamous species ($N = 20$; including polygynous and polygynandrous species). In strictly monogamous species the variance in reproductive success is expected to be identical for males and females. However, strict monogamy is rather rare in animals (*Lukas and Clutton-Brock, 2013*) and most of our sampled monogamous species are birds, which are described as socially monogamous rather than genetically monogamous because all of them show at least some degree of extrapair paternity (*Brouwer and Griffith, 2019*). Moreover, the occurrence of partial polygyny in some sampled bird species rendered their classification problematic. In these problematic cases, we searched the literature for estimates of the proportion of polygynous males in the studied population and only considered those species with <10 % of polygynous males as socially monogamous (*Supplementary file 10*). Hence, even though sexual selection is likely to operate also in most socially monogamous species, we assume that the strength of pre- and postcopulatory sexual selection is generally higher in polygamous compared to monogamous species (*Shuster and Wade, 2003*). We examined

the sensitivity of our classification criteria to different thresholds that have been previously used by other authors and found that the identity of monogamous and polygamous species remained when applying a 5 % (*Moller, 2008*) or a 15 % threshold (*Dunn et al., 2001*).

It is important to stress that our analysis on the relationship between mating system (as a proxy for sexual selection) and the sex difference in genetic variances is rather exploratory and must be considered with caution for at least two reasons. First, our classification of the mating system is clearly an oversimplification of a continuum in the strength of sexual selection across species in nature. Moreover, it is questionable whether the propensity of having multiple mates (i.e., the key characteristic to define the mating system of a given species) translates into competition for mating partners and/or their gametes, which is the essence of sexual selection (*Shuker, 2010*; *Shuker and Kvarnemo, 2021*). Second, in our dataset we found that monogamous species are largely underrepresented (i.e., only 6 out of 26 species) and restricted to vertebrates with only 3 independent evolutionary transitions, which clearly imposes strong limitations on the interpretability of the obtained results (*Uyeda et al., 2018*).

## Statistical analyses

Statistical analyses were carried out in two steps. First, we examined the key assumption of the 'phenotypic gambit' by testing whether estimates of phenotypic variance predict the estimated genetic variance. For this we ran linear regressions with $CV_P$ defined as predictor variable and $CV_G$ defined as response variable. This was done separately for both sexes and the two fitness components. The analyses on the phenotypic gambit were motivated from a methodological perspective and we did not expect that interspecific variation in the difference between $CV_P$ and $CV_G$ can be explained by a shared phylogenetic history. However, for completeness, we also ran linear regressions on PICs computed using the *crunch* function of the *caper* R-package (version 1.0.1) in R (*Orme et al., 2018*) to test whether our findings were robust when accounting for potential phylogenetic nonindependence.

Second, we tested the hypothesis that net selection is stronger on males by testing for a male bias in $CV_P$ and $CV_G$. Specifically, we ran phylogenetic general linear mixed-effects models (PGLMMs) with $CV_P$ or $CV_G$ as the response variable, and sex as a fixed effect. To account for the paired data structure, we added an observation identifier as a random effect. Moreover, all models included a study identifier and the phylogeny (transformed into a correlation matrix) as random effects to account for statistical nonindependence arising from shared study design or phylogenetic history, respectively. Note that the latter also accounts for the nonindependence of estimates obtained from the same species as some studies estimated genetic variances from distinct field populations (*Fox et al., 2004*) or different experimental treatments under laboratory conditions such as food stress (*Holman and Jacomb, 2017*) and temperature stress (*Berger et al., 2014*). In an additional series of PGLMMs, we tested whether our proxy of sexual selection explained interspecific variation in the sex differences of $CV_P$ or $CV_G$ by adding mating system and its interaction with sex as fixed effects to the models. These analyses were run on both the complete dataset and on a subset including only vertebrates. The latter set of analyses was done to acknowledge the fact that all sampled invertebrate species were classified as polygamous and the three independent evolutionary events in our phylogeny marking transitions between mating systems occurred in vertebrates (including six polygamous species and six monogamous species).

Finally, given that primary studies varied in several methodological aspects (see 'Literature search and characterization of primary studies'), we tested whether the different approaches predicted estimates of $CV_P$ and $CV_G$ and affected their sex difference whenever the level of replication allowed statistical analysis. Specifically, we ran three PGLMMs to test separately for effects of the study type, the estimate of genetic variance $V_G$, and the type of reproductive success metric (see above).

We carried out PGLMMs with the *MCMCglmm* R-package (version 2.2.9) (*Hadfield, 2010*), using uninformative priors ($V = 1$, nu = 0.01) and an effective sample size of 20,000 (number of iterations = 11,000,000, burn-in = 1,000,000, thinning interval = 500). We computed the proportion of variance explained by fixed factors ('marginal $R^2$') (*Nakagawa et al., 2013*). In addition, we quantified the phylogenetic signal as the phylogenetic heritability $H^2$ (i.e., proportional variance in $CV_P$ or $CV_G$ explained by the phylogeny) (*de Villemereuil and Nakagawa, 2014*).

In a previous study, testing for sex-specific phenotypic variances in reproductive success (*Janicke et al., 2016*), we ran formal meta-analyses using lnCVR as the tested effect size (*Nakagawa et al.,*

*2014*). This is potentially a more powerful approach for comparing phenotypic variances but rendered unsuitable when comparing genetic variances. This is because the computation of the sampling variance of lnCVR is a function of the sample size of the sampled population and the point estimate of lnCVR (*Nakagawa et al., 2014*). However, genetic variances are estimates from statistical models and notorious for being estimated with low precision (i.e., have large confidence intervals). Therefore, using a meta-analytic approach for genetic variances using lnCVR as an effect size leads to overconfident estimation of the global effect size and is therefore likely to result in type-II-errors. However, to allow comparison with the previous meta-analysis, we report the outcome of phylogenetic meta-analyses on phenotypic variances using lnCVR in *Supplementary file 11*, which largely reflects the results on the point estimates of $CV_P$ from PGLMMs.

## Acknowledgements

We are very grateful to all authors of the primary studies and in particular those providing additional information and/or data including Jessica Abbott, Mats Björklund, Sandra Bouwhuis, Ryan G Calsbeek, Julie Collet, David Hosken, Zenobia Lewis, Jacob Moorad, Tom Tregenza, and Felix Zajitschek. Moreover, we thank Patrice David, Shinichi Nakagawa, Lucas-Marie Orleach, Klaus Reinhardt, Holger Schielzeth, Céline Teplitsky, and Pierre de Villemereuil for discussions and statistical advice. Luke Holeman and two anonymous reviewers provided very constructive comments on a previous draft. Funding: LW and TJ were funded by the German Research Foundation (DFG grant number: JA 2653/2-1). TJ received funds from the Centre national de la recherche scientifique (CNRS). MM was funded by a Marie Curie Individual Fellowship (PLASTIC TERN; grant agreement number: 793550). EHM was funded by a Royal Society University Research Fellowship and the Swedish Research Council (grant number: 2019-03567).

## Additional information

### Funding

| Funder | Grant reference number | Author |
| --- | --- | --- |
| Deutsche Forschungsgemeinschaft | JA 2653/2-1 | Tim Janicke Lennart Winkler |
| H2020 Marie Skłodowska-Curie Actions | 793550 | Maria Moiron |
| Vetenskapsrådet | 2019-03567 | Ted Morrow |

The funders had no role in study design, data collection and interpretation, or the decision to submit the work for publication.

### Author contributions

Lennart Winkler, Data curation, Formal analysis, Investigation, Methodology, Visualization, Writing – original draft; Maria Moiron, Formal analysis, Investigation, Writing – review and editing; Edward H Morrow, Conceptualization, Investigation, Writing – review and editing; Tim Janicke, Conceptualization, Data curation, Formal analysis, Funding acquisition, Investigation, Methodology, Project administration, Supervision, Validation, Visualization, Writing – original draft

### Author ORCIDs

Lennart Winkler http://orcid.org/0000-0003-3597-6540
Maria Moiron http://orcid.org/0000-0003-0991-1460
Edward H Morrow http://orcid.org/0000-0002-1853-7469
Tim Janicke http://orcid.org/0000-0002-1453-6813

### Decision letter and Author response

Decision letter https://doi.org/10.7554/eLife.68316.sa1
Author response https://doi.org/10.7554/eLife.68316.sa2

## Additional files

### Supplementary files
• Supplementary file 1. Results of phylogenetic general linear mixed-effects models (PGLMMs) testing for sex by mating system interaction on phenotypic ($CV_P$) and genetic ($CV_G$) coefficient of variation.

• Supplementary file 2. Phylogenetic general linear mixed-effect models testing for an effect of sex on phenotypic ($CV_P$) and genetic ($CV_G$) coefficient of variation for a reduced dataset including only vertebrates.

• Supplementary file 3. Results of phylogenetic general linear mixed-effects models (PGLMMs) testing for sex by mating system interaction on phenotypic ($CV_P$) and genetic ($CV_G$) coefficient of variation obtained from a reduced dataset including only vertebrates.

• Supplementary file 4. Results of phylogenetic general linear mixed-effects models (PGLMMs) testing for the effect of sex, study type (lab versus field studies) and their interaction on phenotypic ($CV_P$) and genetic ($CV_G$) coefficients of variation.

• Supplementary file 5. Results of phylogenetic general linear mixed-effects models (PGLMMs) testing for the effect of sex, $V_G$ estimate (additive versus total) and their interaction on genetic ($CV_G$) coefficient of variation.

• Supplementary file 6. Results of phylogenetic general linear mixed-effects models (PGLMMs) testing for the effect of sex, $RS$ estimate (temporal versus lifetime reproductive success) and their interaction on phenotypic ($CV_P$) and genetic ($CV_G$) coefficients of variation.

• Supplementary file 7. Analysis of cross-sex genetic correlations.

• Supplementary file 8. Search terms and list of primary studies.

• Supplementary file 9. Methodological characteristics of the 55 primary studies.

• Supplementary file 10. Mating system classification of the 26 sampled species.

• Supplementary file 11. Phylogenetically independent meta-analysis of lnCVR of phenotypic variation.

• Transparent reporting form

### Data availability
All data associated with this study can be retrieved from the Zenodo data repository (https://zenodo.org/record/5529490#.YVFhNKB8Kqc) and R scripts can be accessed at https://lennartwinkler.github.io/Net_Selection_eLife_code/ and the GitHub repository at: https://github.com/LennartWinkler/Net_Selection_eLife_code copy archived at swh:1:rev:5dcb7b744d879bc1f80af8b3dff8184577f005bc.

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
