## [Editor Report]

This study addresses an interesting and important question in evolutionary biology: how does the variance in fitness (components) vary between the sexes? In particular, it aims to evaluate whether there is a larger sex difference in systems with strong sexual selection. This study will be of considerable interest to researchers working on sexual coevolution and the role of sexual selection in promoting adaptation. However, there are some concerns regarding the limitations of the data and methods in support of the conclusions.

---

## [Decision Letter]

**Decision letter after peer review:**

Thank you for submitting your article "Stronger net selection on males across animals" for consideration by *eLife*. Your article has been reviewed by 3 peer reviewers, and the evaluation has been overseen by a Reviewing Editor and Patricia Wittkopp as the Senior Editor. The following individual involved in review of your submission has agreed to reveal their identity: Luke Holman (Reviewer #3).

Essential revisions:

1. In light of the concerns raised below about the phylogenetic non-independence of the species sampled to compare monogamous and polygamous species, we encourage the authors to refocus the manuscript on the different coefficients of variation for fitness between males and females. We suspect that taking the phylogenetic structure of the species included in this analysis into account will significantly weaken the evidence for the conclusions drawn from this analysis.

2. Questions and concerns below about the measure of reproductive success used must be addressed.

3. Reviewer 2's comments about the alternate interpretations in the context of sexual conflict must also be addressed.

In addition to these core issues, the full reviews below also raise a number of additional issues we hope you are able to address.

*Reviewer #1 (Recommendations for the authors):*

My main concerns are outlined in the previous section. A few specific points:

Figure S2 should be in the main text not as supplement. It should also be augmented with two columns on the right side giving the number of estimates for (i) reproductive success and (ii) longevity so it is easy for readers to identify which species are contributing multiple points to Figure 1.

At a minimum, the points in Figure 1 should use different symbols for the major taxonomic groups represented by silhouettes on Figure S2.

*Reviewer #2 (Recommendations for the authors):*

L275: "All PGLMMs were ran with…" Please reword.

L392: I guess this should either be "intra-specific" of "inter-sexual"?

---

## [Author Response]

Essential revisions:1. In light of the concerns raised below about the phylogenetic non-independence of the species sampled to compare monogamous and polygamous species, we encourage the authors to refocus the manuscript on the different coefficients of variation for fitness between males and females. We suspect that taking the phylogenetic structure of the species included in this analysis into account will significantly weaken the evidence for the conclusions drawn from this analysis.

We thank both reviewers for their comments and suggestions regarding the analysis of the social mating system. We agree that this analysis is limited given the very few transitions between monogamy and polygamy in our phylogeny. In the revised version, we toned down the interpretation of our results clarifying that this analysis should be considered preliminary and extensively highlighting its limitations (L 291-302, L 419-433). We also followed reviewer 1’s suggestion to run the analysis on a subset of data including only vertebrates, in which monogamous and polygamous species are more evenly distributed (6 monogamous species, 6 polygamous species). Results of this additional analysis support the previously obtained findings that a male bias in genetic variation of reproductive success can only be found in polygamous species (Table S5 and S6). Nevertheless, the main outcome of our study is the male bias in the genetic variance in reproductive success. The analysis of mating system remains exploratory, which we tried to make clear throughout the revised manuscript.

2. Questions and concerns below about the measure of reproductive success used must be addressed.

In the revised version, we provide a more detailed description of how primary studies differ in several methodological aspects including measurement of reproductive success and paternity analysis (L 203-242). We summarize this heterogeneity in Table S1. Finally, we also show additional analyses testing whether three methodological aspects affect the sex difference in genetic variance. Reassuringly, these analyses suggest that the tested methodological heterogeneity did not bias our findings.

3. Reviewer 2's comments about the alternate interpretations in the context of sexual conflict must also be addressed.

We fully agree with Reviewer 2 that inter- and intra-sexual conflict are essential to predict the role of sexual selection for genome-wide purging of deleterious alleles and its effect on adaptation to changing environments. It is correct that the theory predicting a positive effect of sexual selection on adaptation relies on the assumption that inter- and intra-locus sexual conflict play a minor role and have very limited fitness effects. We followed the reviewer’s suggestions, explaining in detail the importance of sexual conflict (L 48-64) and clarifying that the aim of our study is to test the other assumption of the theoretical framework, which is that net selection is male-biased.

In addition to these core issues, the full reviews below also raise a number of additional issues we hope you are able to address.

We have now implemented all additional comments and suggestions raised by the reviewers.

Reviewer #1 (Recommendations for the authors):My main concerns are outlined in the public review section. A few specific points:Figure S2 should be in the main text not as supplement. It should also be augmented with two columns on the right side giving the number of estimates for (i) reproductive success and (ii) longevity so it is easy for readers to identify which species are contributing multiple points to Figure 1.

Following the recommendations, we moved Figure S2 to the main text and added 2 columns with the sample sizes as suggested (now Figure 1).

At a minimum, the points in Figure 1 should use different symbols for the major taxonomic groups represented by silhouettes on Figure S2.

When revising Figure 1, we initially followed the reviewer suggestion and used different symbols for every major taxonomic group but we got the impression that this made the graphical representation of the observed relationship overloaded and confusing. Therefore, we decided to compromise by differentiating only between vertebrates and invertebrates. Should the reviewer believe otherwise, we would be happy to add them, but at this stage we believe that this solution reflects the suite of analyses presented in the manuscript.

Thank you very much for these inspiring and constructive comments.

Reviewer #2 (Recommendations for the authors):L275: "All PGLMMs were ran with…" Please reword.

We rephrased this sentence.

L392: I guess this should either be "intra-specific" of "inter-sexual"?

Thanks for spotting this error. We changed it to “cross-sex genetic correlation”.